

# Genotypic and phenotypic characterization of thermo-sensitive genic male sterile (TGMS) rice lines using simple sequence repeat (SSR) markers and population structure analysis

B Nagendra Naidu[1], Manonmani Swaminathan[1], Pushpam Ramamoorthy[2], Kumaresan Dharmalingam[1], Raveendran Muthurajan[3], Selvi Duraisamy[4], Nivedha Rakkimuthu[1], Abirami Subramanian[1], Rithesh Natarajan[1] and Bonipas Antony John[1]

[1] Department of Rice, Tamil Nadu Agricultural University, Coimbatore, Tamil Nadu, India
[2] Department of Forage Crops, Tamil Nadu Agricultural University, Coimbatore, Tamil Nadu, India
[3] Directorate of Research, Tamil Nadu Agricultural University, Coimbatore, Tamil Nadu, India
[4] Department SS&AC, Tamil Nadu Agricultural University, Coimbatore, Tamil Nadu, India

## ABSTRACT

Thermo-sensitive genic male sterile (TGMS) lines in rice are crucial for hybrid breeding, enhancing genetic diversity by eliminating the need for manual emasculation and restorer genes. These lines induce sterility at high temperatures and restore fertility at low temperatures, in contrast to cytoplasmic male sterility (CMS) systems that require specific restorative genes. This temperature-sensitive mechanism allows for greater flexibility in pairing parent lines, increasing genetic diversity and enabling recombination of beneficial traits in hybrids. A randomized block design (RBD) with three replications was employed for the evaluation of these TGMS rice lines. This study investigates the molecular diversity and genetic variability among TGMS rice lines. Traits such as single plant yield, grains per panicle, glume angle, and pollen fertility showed significant phenotypic and genotypic variation, indicated by high coefficients of variation (PCV and GCV), heritability estimates, and genetic advance as a percentage of mean (GAM). These results highlight substantial genetic variation and selection potential. Euclidean distance matrix analysis of morphological data revealed notable genetic differences. TNAU 137S 1 and TNAU 137S 2 were the most genetically similar, while TNAU 112S and TNAU 114S showed the greatest divergence. Principal component analysis (PCA) revealed distinct genetic profiles among lines such as TNAU 136S, TNAU 113S, TNAU 142S, and TNAU 126S, important for hybrid development. Molecular diversity analysis using simple sequence repeat (SSR) markers identified 90 alleles and eight genetic clusters. Bayesian analysis further confirmed two major subpopulations with significant genetic divergence. These findings support the selective use of parent lines for hybrid rice breeding.

Corresponding author
Manonmani Swaminathan,
manonmanitnau@gmail.com

# INTRODUCTION

Rice (*Oryza sativa* L.) is a critical cereal crop and the primary food source for over half of the global population, especially in Asia. The FAO forecasts that world rice production will reach a record 535.1 million tonnes in 2024, driven by improved harvests of maize and wheat in key exporters. As the global population grows, the development of rice hybrids with higher yield potential and stable production is crucial for food security (*Checco et al., 2023*).

The cytoplasmic male sterility (CMS) system, also known as the three-line system, has long been used in hybrid rice breeding. This system, which dates back to the early 1970s, includes three components: the male sterile line (A-line), the maintainer line (B-line), and the restorer line (R-line) (*Vanitha et al., 2020*). However, compared to high-yielding varieties (HYVs), the CMS system offers limited yield increases (15–20%) and has constraints on rice germplasm usage (*Azad et al., 2022*). To address these limitations, the thermo-sensitive genic male sterile (TGMS) system offers a more efficient solution by removing the need for maintainer and restorer lines. In this system, temperature-sensitive sterility is induced, allowing for more flexible and diverse parent pairings, which expands the genetic base and enhances hybrid vigor. The TGMS system can also streamline the breeding process by focusing solely on TGMS lines and pollinators, without the need for complex cytoplasmic sterility or restorative gene compatibility.

Unlike CMS, TGMS uses temperature to induce sterility, making it particularly suitable for tropical countries like India, where temperature fluctuations are significant both seasonally and with altitude changes. The TGMS gene can be integrated into any rice variety, offering greater flexibility in hybrid production (*He et al., 2021*). Institutions like International Rice Research Institute (IRRI), Tamil Nadu Agricultural University (TNAU), and International Institute of Rural Reconstruction (IIRR) have been instrumental in developing TGMS lines through mutation breeding and introgression of TGMS donors into various genetic backgrounds. Despite its advantages, the TGMS system requires precise management of sterility during the hybrid seed production process. Proper timing of planting and careful site selection are crucial for maintaining sterility in TGMS rice lines and ensuring the production of high-quality hybrid seeds. Additionally, research into the environmental adaptability of TGMS lines is essential, as their sterility-fertility balance can be affected by environmental stresses such as temperature fluctuations. For instance, TGMS lines with lower critical sterility point (CSP) thresholds, like Type 3, are more stable in temperate or subtropical regions, while those with higher CSP thresholds, like Type 1, are better suited for warmer climates. Understanding these factors is critical for optimizing TGMS lines across diverse agro-climatic zones and ensuring consistent hybrid seed production.

With a high CSP over 32 °C and a low critical fertility point (CFP) below 24 °C, Type 1 TGMS lines are categorized as optimal for hybrid seed production and thermo-sensitive genic male sterile (TGMS) (TGMS) multiplication. Notable examples of lines in this category include GD-1 and PMT-22, which have been identified as effective TGMS lines for regions with high temperatures during the hybrid seed production season (*Chen et al.,*

*2022*). SM3 and SM5 have also been referenced in earlier studies, but recent advancements highlight the increasing potential of lines like GD-1, particularly in controlled environments (*Zhou et al., 2016*). Type 2 TGMS lines, including notable Chinese EGMS lines like the 7001 S type and Pei'ai 64S, exhibit high CSP and CFP thresholds exceeding 32 °C and 24 °C, respectively. These lines are extensively used in China for hybrid seed production in environments with narrow temperature ranges (*Fan & Zhang, 2018*; *Zhou et al., 2016*). The genetic and molecular characterization of such lines has highlighted their adaptability in specific environmental conditions, making them valuable for hybrid rice production (*Neethu, 2016*).

In contrast, all our TGMS lines belong to Type 3, characterized by a low CSP below 32 °C and a low CFP below 24 °C. This type is ideal for hybrid seed production as it provides consistent sterility across diverse environments, though seed multiplication challenges persist. Well-documented examples of Type 3 TGMS lines include JP8-1A-12, SA-2 (F43), and F61, as well as DRR17A and IR58025B, which have shown excellent adaptability in subtropical regions like India (*Chen et al., 2022*; *Peng et al., 2023*). These lines are particularly suited to subtropical conditions due to their reliable sterility expression and high adaptability. Lastly, no discernible lines exist for Type 4, which features a low CSP below 32 °C and a high CFP above 24 °C. This category remains a theoretical framework, as lines with such specific thresholds are yet to be identified or bred (*Fan & Zhang, 2018*; *Peng et al., 2023*). This classification serves as a valuable guideline for selecting TGMS lines suited to specific climates, enabling efficient hybrid seed production across diverse environmental conditions.

The study of molecular markers, particularly simple sequence repeats (SSRs), has enhanced understanding of the genetic diversity among TGMS lines. Genetic diversity is crucial for the resilience of rice crops and their ability to adapt to various environmental challenges (*Pardeep, 2015*). SSR markers, being multi-allelic, co-dominant, and highly polymorphic, are ideal for analyzing genetic diversity across TGMS lines with diverse genetic backgrounds. They provide comprehensive allele frequency data and effectively capture heterozygosity, which is critical for utilizing TGMS in heterosis breeding. In contrast, single nucleotide polymorphisms (SNPs), being bi-allelic, offer limited heterozygosity information, while dominant markers like amplified fragment length polymorphism (AFLP) fail to detect heterozygosity and may vary with experimental conditions, impacting reproducibility. Studies on the inheritance of TGMS and the characterization of new lines for their sterile and fertile phases are vital for the successful application of this male sterility system in heterosis breeding (*Wang et al., 1995*).

Advancements in molecular breeding, such as marker-assisted selection (MAS), have significantly accelerated TGMS line development. MAS enhances the efficiency of breeding by precisely incorporating the TGMS gene into elite rice varieties. For instance, *Ni et al. (2015)* improved the hybrid Liangyou6326 by using MAS to enhance Guangzhan63S with Pi9 and Xa23 genes, improving its performance and disease resistance. *Kavithamani et al. (2024)* identified RM 3476 as associated with the TGMS gene in TS 29, facilitating the selection of male-sterile plants for hybrid seed production. Similarly, *Chen et al. (2024)* improved blast resistance in TGMS line HD9802S by integrating the R6 gene *via* MAS and

anther culture without compromising key traits. Combining traditional breeding methods with advanced molecular techniques enables the development of TGMS lines with superior agronomic traits, including enhanced yield, disease resistance, and stress tolerance (*Zhang et al., 2024*). The availability of diverse TGMS lines facilitates the creation of hybrids with improved resistance, quality, and yield. A deeper understanding of phenotypic diversity allows breeders to select complementary lines with desirable agronomic traits, resulting in superior hybrid performance (*Roy & Kumaresan, 2018*).

# MATERIAL AND METHODS

## Plant materials

During the 2022 *rabi* season, 57 TGMS lines ( Table S1) were evaluated at the Hybrid Rice Evaluation Centre in Gudalur and the Paddy Breeding Station of Tamil Nadu Agricultural University (TNAU) in Coimbatore, India. The Paddy Breeding Station, located at a latitude of 11°N, longitude of 77°E, and an altitude of 426.72 m above sea level, features clayey soil with a pH of 7.8, an annual average temperature of 26 °C, and relative humidity of 73 percent. These stable conditions are ideal for studying the sterility behavior of TGMS lines. The lines were developed through various breeding approaches, including pedigree breeding, spontaneous mutation, and selection, with most derived from crosses between indica and japonica subspecies to incorporate traits from both genetic backgrounds, enhancing adaptability and efficacy for hybrid seed production.

A randomized block design (RBD) with three replications was employed for the evaluation. Each block consisted of two rows per genotype, ensuring a randomized and balanced layout across the experimental plots. Thirty-day-old seedlings were transplanted at a spacing of 20 × 20 cm, following standard recommendations for fertilizer application, irrigation, and pest management as per the TNAU Agritech Portal (https://agritech.tnau.ac.in). Observations were recorded from five individual plants per replication, amounting to a total of 15 plants per genotype (five plants × three replications). Data were collected for 13 agronomic and floral traits: days to 50% flowering (DFF), plant height (PH), number of productive tillers per plant (PT), panicle exertion (PE), panicle length (PL), pollen fertility (PF), spikelet fertility (SF), stigma length (SL), stigma exertion (SE), glume angle (GA), number of grains per panicle (GPP), and single plant yield (SPY). This detailed experimental setup ensured the generation of robust and reliable data for evaluating the performance of the TGMS lines.

**Panicle exertion percentage (PE):** The Panicle Exertion Percentage was calculated using the following formula, which considered the exerted length of the panicle relative to the total panicle length (*Kumar & Manohar, 2020*).

$$\text{Panicle exertion percentage} = (\text{Panicle length exerted/Total panicle length}) \times 100.$$

**Spikelet fertility (SF):** The spikelet fertility was calculated using the following formula, which considered the number of fertile spikelets relative to the total spikelets.

$$\text{Spikelet Fertility}(\%) = (\text{Number of Fertile Spikelets/Total Number of Spikelets}) \times 100.$$

**Stigma exertion percentage (SE):** Stigma exertion percentage was calculated based on the number of spikelets with exerted stigma relative to the total number of spikelets (*Vinodhini et al., 2019*).

Stigmas exertion percentage = (number of spikelets with stigma exertion/ total number of spikelets) × 100.

## Molecular characterization using SSR markers

Genomic DNA was extracted from leaf samples of 3–4-week-old seedlings using the *Doyle & Doyle (1987)* protocol. The purity of the isolated DNA was assessed using a Genesys UV spectrophotometer (Genesys, Menlo Park, CA, USA) measuring optical density at 260 and 280 nm, with an OD260/280 ratio of approximately 1.8 deemed indicative of pure DNA. Polymerase chain reaction (PCR) was conducted with a reaction mixture containing two µl of template DNA (20 ng/µl), 0.5 µl each of forward and reverse primers (10 µM), four µl of 2X BioServe mastermix (comprising DNA polymerase (0.5–1.25 U), dNTPs (200 µM each), 1X buffer, $MgCl_2$ (1.5–2.5 mM), and nuclease-free water), and three µl of sterile water. SSR analysis utilized 43 primers selected for their polymorphism to represent all rice chromosomes except chromosome 6. PCR cycling conditions included an initial denaturation at 95 °C for 5 min, followed by 35 cycles of 30 s at 94 °C, 30 s at 55 °C (or 50 °C for some primers), and 30 s at 72 °C, with a final extension at 72 °C for 10 min. The amplified products were resolved on a 3% agarose gel in 1X TBE buffer alongside a 100 bp ladder (Bio-Helix) and visualized using Bio-Rad imaging equipment. Amplified products were scored based on the presence ('1') or absence ('0') of distinct, well-defined bands of expected sizes, as independently confirmed by two researchers (*Lavudya et al., 2024*). Non-specific amplifications, weak bands, and artefacts were excluded from analysis. PCR optimization, involving adjustments to annealing temperature, cycle numbers, and DNA concentrations, was performed for weak amplifications to ensure reproducibility. Molecular cluster analysis was conducted using the Jaccard distance method (*Jaccard, 1901*) based on the binary scoring of alleles detected with the 43 polymorphic SSR primers. Details of the amplified base pairs and polymorphic markers are presented in Table S2.

### Statistical analysis

The analysis of variance (ANOVA), principal component analysis (PCA), and assessments of genetic variability were conducted using R Studio version 4.2.3, employing the packages "variability", "Agricolae", "FactoMineR", and "factoextra". Genetic advance as a percentage of mean (GAM) was calculated using the formula GAM (%) = (K × $\sigma_x/\mu$) ×100, where K is the selection differential (2.06 for 5% selection intensity), $\sigma_x$ is the genotypic standard deviation, and µ is the population mean, to estimate potential genetic improvement for selected traits in breeding programs. For further data exploration, PCA combined with the Multi-Trait Genotype-Ideotype Distance Index (MGIDI), using the Metan and Stats packages in R (*R Core Team, 2023*), facilitated multi-trait selection. Standardized trait data, weighted by importance, were used to calculate Euclidean distances from the ideal genotype (ideotype). Lower MGIDI scores indicated genotypes with superior
**Table 1  ANOVA for all the biometrical traits.**

| Source of variation | df | DFF | PH | PT | PE | PL | PF (%) | SF (%) | SL (mm) | SE (%) | GA | GPP | SPY (g) |
|---|---|---|---|---|---|---|---|---|---|---|---|---|---|
| Genotype | 56 | 650.90** | 163.58** | 35.37** | 77.79** | 17.88** | 888.83** | 196.62** | 0.198** | 368.57** | 117.44** | 8,350.30** | 234.51** |
| Replication | 2 | 6.01 | 7.83 | 2.09 | 1.81 | 0.51 | 13.93 | 6.43 | 0.003 | 1.08 | 1.66 | 15.90 | 0.643 |
| Error | 112 | 28.47 | 21.25 | 1.05 | 12.74 | 1.39 | 15.05 | 8.44 | 0.010 | 5.25 | 1.12 | 81.40 | 0.632 |

Notes.

The traits measured include DFF, Days to 50% Flowering; PH, Plant Height in cm; PT, Productive Tillers per Plant; PE, Panicle Exertion in cm; PL, Panicle Length in cm; PF, Pollen Fertility %; SF, Spikelet Fertility %; SL, Stigma Length in mm; SE, Stigma Exertion %; GA Glume Angle in degrees; GPP, Grains per Panicle; and SPY, Single Plant Yield in g, with double asterisks (**) indicating a highly significant difference at the 1% level ($p < 0.01$).

multi-trait performance. Molecular clustering was based on marker scores, utilizing Jaccard distance, heterozygosity index, and polymorphism information content (PIC) values, which were analyzed using the R-shiny-based software "PBPERFECT" (*Allan, 2023*). Population structure analysis was carried out with STRUCTURE 2.3.4, a Bayesian model-based method incorporating 50,000 burn-in iterations, with results visualized through STRUCTURE HARVESTER. Additional metrics of genetic diversity, such as genetic differentiation (Fst), expected heterozygosity (He), observed heterozygosity (Ho), and analysis of molecular variance (AMOVA), were calculated using GenAlex version 6.5 (*Peakall & Smouse, 2007*).

# RESULTS AND DISCUSSION

## Genetic variability

The ANOVA results reveal highly significant genotype effects ($p < 0.01$) for all biometric traits, indicating substantial genetic variation among the genotypes (Table 1). Minimal replication effects suggest limited environmental influence on the traits measured, supporting the reliability of genetic selection for breeding programs.

Among the evaluated traits, plant height, productive tillers, panicle length, and single plant yield emerged as critical contributors to hybrid rice breeding. Single plant yield (Table 2) demonstrated high genetic variability and heritability (PCV: 35.287%, GCV: 34.864%, H: 98%) (*Ashraf et al., 2020*), making it a key target for hybridization programs due to its robust genetic control and minimal environmental dependency. Similarly, plant height (PCV: 10.115%, GCV: 8.405%, H: 69%) plays a vital role in ensuring compact plant architecture, contributing to lodging resistance and improved adaptability under dense planting conditions (*Pillai, Srimathi & Aananthi, 2019*). Productive tillers (PCV: 19.735%, GCV: 18.888%, H: 92%) require a balanced approach combining genetic selection and environmental management to maximize their potential (*Roy & Kumaresan, 2019*). Panicle length (PCV: 12.067%, GCV: 10.774%, H: 80%) remains a significant trait for optimizing yield-related characteristics in hybrid rice despite some environmental influence (*Kavithamani et al., 2013*). Focusing on these traits, the study underscores their importance in improving reproductive efficiency, yield, and adaptability in hybrid rice. These findings align with molecular and genetic strategies to optimize TGMS lines, providing a foundation for reliable progress in rice breeding programs (*Salgotra, Gupta & Ahmed, 2012*).

**Table 2  Genetic variability, heritability and genetic advance in TGMS rice lines.**

|  | PCV | Range | GCV | Range | Heritability (%) | Range | GAM(%) | Range |
|---|---|---|---|---|---|---|---|---|
| Days to fifty percent flowering | 14.406 | Moderate | 13.509 | Moderate | 88 | High | 26.095 | High |
| Plant height | 10.115 | Moderate | 8.405 | Low | 69 | High | 14.389 | Moderate |
| Number of productive tillers | 19.735 | Moderate | 18.888 | Moderate | 92 | High | 37.238 | High |
| Panicle exertion | 9.266 | Low | 7.354 | Low | 63 | High | 12.025 | Moderate |
| Panicle length | 12.067 | Moderate | 10.774 | Moderate | 80 | High | 19.817 | Moderate |
| Pollen fertility | 26.057 | High | 25.409 | High | 95 | High | 51.042 | High |
| Spikelet fertility | 15.022 | Moderate | 14.103 | Moderate | 88 | High | 27.275 | High |
| Stigme length | 14.364 | Moderate | 13.344 | Moderate | 86 | High | 25.536 | High |
| Stigma exertion | 27.899 | High | 27.313 | High | 96 | High | 55.084 | High |
| Glume angle | 31.508 | High | 31.059 | High | 97 | High | 63.072 | High |
| Grains per panicle | 32.240 | High | 31.774 | High | 97 | High | 64.509 | High |
| Single plant yield | 35.287 | High | 34.864 | High | 98 | High | 70.962 | High |

**Notes.**
PCV (%), Phenotypic Coefficient of Variation; GCV (%), Genotypic Coefficient of Variation; GAM (%), Genetic Advance as a Percentage of Mean.

## Morphological diversity

The morphological diversity observed in TGMS rice lines is critical for two-line hybrid rice breeding, offering opportunities to exploit heterosis from genetically distant parents. Significant Euclidean distances between pairs like TNAU 112S and TNAU 114S (308.51) and TNAU 112S and TNAU 113S (296.97) (Fig. S1) indicate the potential to develop high-yield heterotic hybrids. Enhanced hybrid vigour, a primary goal in hybrid rice development, is achieved through the crossing of such genetically diverse lines. Additionally, the genetic distinctiveness of lines such as TNAU 38S and TNAU 114S underscores the importance of diversity in hybrid breeding. Significant genetic variation not only increases yield potential but also enhances the stress resilience of hybrid progeny. Conversely, minimal variation, as seen in closely related genotypes like TNAU 137S 1 and TNAU 137S 2 (4.03), offers limited benefits for hybrid vigour. This observation is consistent with the findings of *Barman et al. (2019)*, who noted that limited genetic divergence provides restricted advantages in hybrid breeding. Their research highlights that higher hybrid vigour is directly associated with the crossing of genetically distinct lines, further validating the importance of selecting diverse TGMS lines to maximize hybrid productivity and resilience. These results, supported by *Barman et al. (2019)*, underscore the indispensable role of genetic and morphological diversity in advancing two-line hybrid rice breeding programs, confirming the strong correlation between genetic divergence and hybrid rice productivity.

## Principal component analysis

PCA provides a comprehensive understanding of the variability and underlying structure of the evaluated traits, as shown in Table 3. The analysis identified twelve principal components (PCs), with the first five components explaining a significant portion of the total variance 65.261%. The first principal component (PC1) accounted for 17.267% of

**Table 3** The factor loadings, eigen values, percent of variance and cumulative percent of variance for all principal components.

| Variable | PC1 | PC2 | PC3 | PC4 | PC5 | PC6 | PC7 | PC8 | PC9 | PC10 | PC11 | PC12 |
|---|---|---|---|---|---|---|---|---|---|---|---|---|
| DF | −0.214 | −0.066 | −0.613 | −0.136 | −0.195 | 0.249 | 0.047 | −0.386 | −0.223 | 0.035 | −0.218 | −0.454 |
| PH | −0.488 | 0.121 | 0.262 | −0.201 | −0.221 | 0.052 | 0.193 | −0.023 | 0.293 | −0.319 | −0.571 | 0.187 |
| PT | 0.407 | 0.184 | −0.205 | −0.114 | −0.183 | 0.153 | 0.648 | 0.109 | −0.225 | 0.145 | −0.126 | 0.411 |
| PE | 0.343 | 0.446 | 0.042 | 0.036 | 0.092 | 0.175 | −0.143 | −0.187 | −0.181 | −0.738 | 0.041 | −0.082 |
| PL | −0.458 | 0.310 | −0.079 | 0.165 | −0.296 | −0.026 | 0.263 | −0.119 | 0.079 | −0.079 | 0.684 | 0.097 |
| PF | −0.353 | 0.313 | −0.150 | 0.103 | 0.315 | −0.420 | −0.138 | 0.047 | −0.577 | 0.078 | −0.201 | 0.259 |
| SF | 0.056 | 0.452 | 0.112 | −0.287 | −0.443 | −0.067 | −0.218 | 0.503 | −0.105 | 0.203 | −0.019 | −0.378 |
| SL | −0.052 | −0.150 | 0.381 | −0.571 | −0.115 | 0.204 | −0.205 | −0.365 | −0.386 | 0.124 | 0.212 | 0.252 |
| SE | 0.046 | −0.238 | 0.468 | 0.353 | −0.258 | −0.267 | 0.356 | −0.141 | −0.405 | −0.072 | −0.085 | −0.370 |
| GA | 0.017 | 0.356 | 0.197 | −0.315 | 0.523 | −0.134 | 0.374 | −0.266 | 0.204 | 0.240 | 0.036 | −0.363 |
| GPP | −0.112 | −0.379 | −0.171 | −0.463 | 0.176 | −0.225 | 0.250 | 0.423 | −0.133 | −0.442 | 0.217 | −0.128 |
| SPY | 0.278 | −0.006 | −0.195 | −0.206 | −0.321 | −0.720 | −0.114 | −0.362 | 0.240 | −0.065 | −0.022 | 0.121 |
| Eigen value | 2.072 | 1.785 | 1.522 | 1.317 | 1.135 | 0.912 | 0.849 | 0.656 | 0.572 | 0.546 | 0.365 | 0.268 |
| % variance | 17.267 | 14.874 | 12.686 | 10.978 | 9.459 | 7.600 | 7.077 | 5.466 | 4.768 | 4.551 | 3.043 | 2.233 |
| Cumulative % | 17.267 | 32.140 | 44.826 | 55.804 | 65.262 | 72.863 | 79.940 | 85.406 | 90.174 | 94.724 | 97.767 | 100.000 |

**Notes.**

DFF, Days to 50% flowering; PH, Plant height (cm); PT, productive tillers per plant; PE, Panicle exertion (cm); PL, Panicle length (cm); PF, Pollen fertility (%); SF, Spikelet fertility (%); SL, Stigma length (mm); SE, Stigmaexertion (%); GE, Glume angle (°); GPP, Grains per panicle; SPY, Single plant yield (g).

the variance, and the second principal component (PC2) explained 14.874% (Fig. 1). The eigenvalues for the first five components, all greater than one, indicated their considerable contribution to the dataset's variance: PC1 (2.072), PC2 (1.785), PC3 (1.522), PC4 (1.317), and PC5 (1.135).

Loadings from the PCA revealed key relationships between traits and components. PC1 showed an inverse relationship with plant height and pollen fertility, with loadings of −0.488 and −0.353, respectively, aligning with findings from *Singh et al. (2024)*. PC2 exhibited positive loadings for panicle exertion (0.446) and spikelet fertility (0.452), suggesting its role in distinguishing these traits (*Vanisri et al., 2020*). Traits with negative loadings on PC2, such as grains per panicle (−0.379) and days to 50% flowering (−0.066), showed an inverse relationship with this component. In PC3, stigma exertion (0.468) and productive tillers (0.648) had significant positive loadings, indicating their major contribution, while panicle length (−0.296) and spikelet fertility (0.112) were less influential. The wide distribution of genotypes in the PCA biplot (Fig. 2) revealed substantial genetic diversity among the TGMS lines. Key agronomic traits, including spikelet fertility (SF), panicle exertion (PE), number of grains per panicle (NGP), plant height (PH), panicle length (PL), and number of productive tillers (PT), played a significant role in the observed variation. The vector length of each trait in the biplot indicates its contribution to the overall variation (Fig. 3). Notably, TNAU 16S and TNAU 136S, located far from other genotypes, exhibited high productive tillers (PT) and panicle exertion (PE), making them potential candidates for enhancing yield-related traits.

TNAU 113S and TNAU 92S, located close to each other, shared traits related to pollen fertility (PF), indicating their suitability for studying sterility mechanisms. Meanwhile,

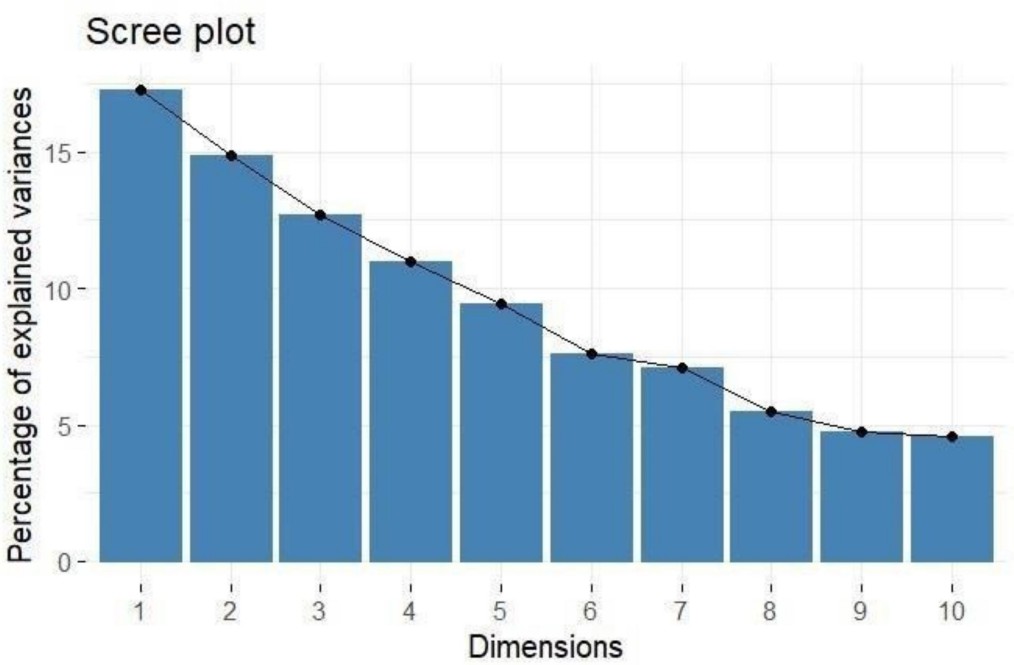

**Figure 1** Scree plot showing percentage of explained variance.

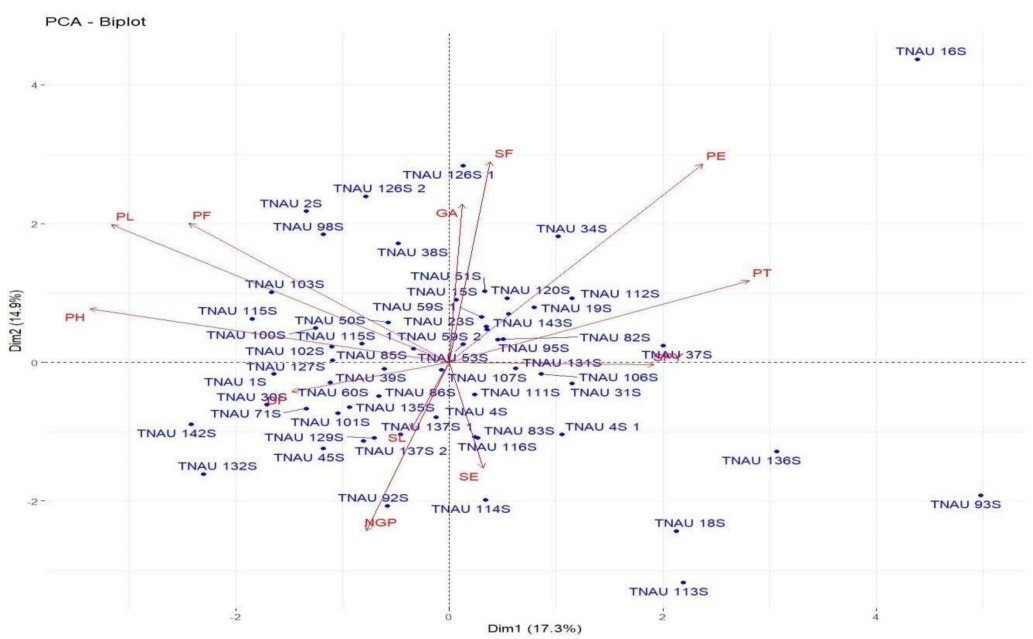

**Figure 2** Principal component analysis (PCA) biplot depicting genotype distribution and diversity.

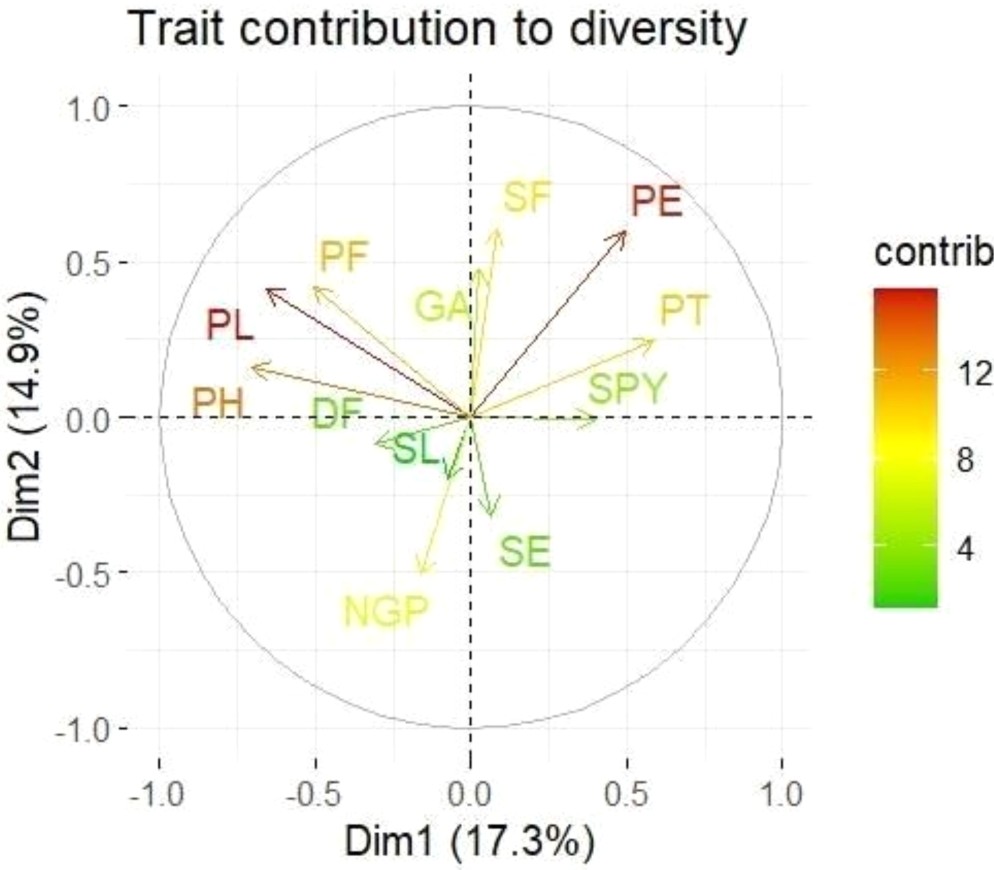

**Figure 3** Principal component analysis (PCA) biplot showing trait contributions to diversity.

TNAU 137S-1 and TNAU 137S-2 displayed balanced traits, particularly spikelet fertility (SF) and stigma exertion (SE), making them valuable for genetic studies. Additionally, TNAU 18S and TNAU 126S-1, aligned with grain yield (GP) and single plant yield (SPY), emerged as promising genotypes for yield improvement, particularly in stress-prone environments. These findings align with *Ramakrishnan et al. (2016)*, who highlighted PCA's utility in identifying genotypes with traits contributing most to yield under stress conditions.

Furthermore, insights from *Girgel (2021)* corroborate the significance of PCA in differentiating genotypes based on agronomic traits, underscoring its role in stress resilience analysis. Similarly, *Mustafa et al. (2015)* emphasized PCA's effectiveness in identifying maize genotypes resilient to water stress, echoing the relevance of spikelet fertility and stigma exertion as pivotal traits. Together, these studies strengthen the conclusion that PCA is a powerful tool for uncovering key trait relationships and genetic diversity, paving the way for the selection and improvement of genotypes with stress-tolerant and yield-enhancing characteristics.

**Table 4  Estimates of selection differential, selection gain, and heritability based on MTSI for thirteen biometrical traits.**

| Variable | Factor | Xo | Xs | SD | H2 | SG |
|---|---|---|---|---|---|---|
| PH | FA1 | 81.9 | 79.9 | −2.01 | 0.87 | −2.14 |
| PL | FA1 | 21.8 | 20.8 | −1 | 0.922 | −4.26 |
| PF | FA1 | 67.2 | 70.7 | 3.56 | 0.983 | 5.21 |
| PT | FA2 | 17.9 | 20.2 | 2.32 | 0.97 | 12.6 |
| SF | FA2 | 56.2 | 60.5 | 4.33 | 0.957 | 7.37 |
| SPY | FA2 | 25 | 31.7 | 6.71 | 0.992 | 26.7 |
| DFF | FA3 | 107 | 103 | −4.1 | 0.956 | −3.68 |
| SE | FA3 | 40.3 | 40.1 | −0.178 | 0.986 | −0.434 |
| SL | FA4 | 1.88 | 1.91 | 0.0266 | 0.95 | 1.34 |
| NGP | FA4 | 165 | 148 | −17.2 | 0.99 | −103 |
| PE | FA5 | 63.3 | 67.5 | 4.17 | 0.836 | 5.50 |
| GA | FA5 | 20 | 26.6 | 6.56 | 0.99 | 32.4 |

Notes.

Xo, overall mean of genotypes; Xs, mean of the selected genotypes; SD, selection differential; SG, selection gain or impact; h2, heritability; DFF, Days to 50% flowering; PH, Plant height (cm); PT, productive tillers per plant; PE, Panicle exertion (cm); PL, Panicle length (cm); PF, Pollen fertility (%); SF, Spikelet fertility (%); SL, Stigma length (mm); SE, Stigmaexertion (%); GE, Glume angle (°); GPP, Grains per panicle; SPY, Single plant yield (g).

## Multi-trait genotype–ideotype distance index for identifying superior TGMS lines

MGIDI analysis for TGMS rice lines highlights a logical approach toward optimizing traits crucial for hybrid rice breeding. This strategy focuses on structural, reproductive, and flowering traits, ensuring genetic stability and high performance. Shorter plant stature, a central objective in rice breeding programs, has been repeatedly emphasized for its role in improving stand ability and lodging resistance under dense planting conditions (*Khush, 2001*). This aligns with the observed negative selection differential for plant height (SD = −2.01) and panicle length (SD = −1.00), underscoring the importance of compact architecture for hybrid rice systems (Table 4). The high heritability values for these traits ($H^2 = 0.87$ for plant height and $H^2 = 0.922$ for panicle length) confirm their genetic stability across generations. According to *Khush (2001)*, the success of the Green Revolution relied heavily on incorporating such high-yielding, lodging-resistant traits into breeding programs, and the current selection trends in TGMS rice lines are a testament to the continued relevance of this approach.

Reproductive efficiency traits, such as pollen fertility (PF) and spikelet fertility (SF), demonstrate significant positive selection gains (SG = 5.21 for PF and SG = 7.37 for SF), coupled with exceptionally high heritability ($H^2 = 0.983$ and $H^2 = 0.957$, respectively). *Virmani & Kumar (1994)* highlighted the importance of these traits for hybrid seed production, as they directly influence the seed setting rate and overall hybrid vigor. The positive selection differential for productive tillers (SD = 2.32) and its high heritability ($H^2 = 0.97$) underscore the trait's critical role in enhancing yield. These findings align with *Virmani & Kumar (1994)* assertion that optimizing productive tillers and other yield-contributing traits is vital for stable hybrid performance.

Single plant yield (SPY), with a remarkable heritability value ($H^2 = 0.992$) and the largest selection gain (SG = 26.7), further underscores the potential for yield improvement in TGMS lines. This result resonates with *Peng, Khush & Cassman (2008)*, who emphasized the challenge of enhancing yield potential in tropical rice. They noted that while IR8 set a high benchmark for yield potential, continued genetic improvements, as seen in the TGMS lines, are essential for meeting future productivity demands. Interestingly, the negative selection differential for grains per panicle (SD = −17.2) suggests a trade-off that prioritizes other yield components, such as grain filling and uniformity, as also discussed by *Peng, Khush & Cassman (2008)*. Flowering traits like days to 50% flowering (DFF), stigma exertion (SE), glume angle (GA), and stigma length (SL) play a key role in improving pollination success and hybridization efficiency. DFF ensures flowering synchronization between male and female plants, while SE, GA, and SL enhance stigma accessibility and pollen capture. Together, these traits optimize hybridization, leading to higher hybrid seed yield and quality. Earlier flowering, indicated by a negative selection differential (SD = −4.1), enhances synchronization of blooming times, a critical factor for successful hybrid seed production. High heritability values for flowering traits, including stigma exertion ($H^2 = 0.986$) and glume angle ($H^2 = 0.99$), ensure their consistent expression across generations. These results further validate the conclusions of *Khush (2001)*, who emphasized that advancements in plant breeding must integrate traits that optimize resource use and reproductive success.

The selected TGMS lines as shown in the Fig. 4 *viz.*, TNAU 106S, TNAU 37S, TNAU 143S, TNAU 51S, TNAU 136S, TNAU 19S, TNAU 126S-2, TNAU 16S, and TNAU 126S-1 exhibit a combination of high heritability for key traits such as SPY, GA, PE, and PF, as well as positive selection trends for reproductive and flowering efficiency. These findings validate the utility of MTSI in identifying and improving critical traits, ensuring the genetic robustness and adaptability of TGMS lines in hybrid rice breeding programs. The additional references, particularly *Khush (2001)*, *Peng, Khush & Cassman (2008)*, and *Virmani & Kumar (1994)*, reinforce the relevance of these results, demonstrating the alignment of current breeding strategies with established principles of hybrid rice development. Together, these studies provide a comprehensive genetic foundation for improving hybrid rice production, ensuring performance stability and yield enhancement in diverse growing environments.

## Molecular diversity
### Unweighted pair group method with arithmetic mean (UPGMA) clustering
The genetic diversity analysis of the 57 parental lines using 43 SSR markers revealed a total of 90 alleles, with the number of alleles per marker ranging from 2 to 3. *Sai Rekha et al. (2021)* observed an average of 2.67 polymorphic alleles per marker, consistent with the findings of this study. *Singh et al. (2011)*, however, reported a slightly higher average of 2.76 alleles per marker in rice, indicative of greater allelic diversity in their study populations, which were likely more genetically diverse than the TGMS parental lines in this analysis. The polymorphic information content (PIC) and heterozygosity index (H) are critical metrics in evaluating genetic diversity and marker effectiveness in rice breeding studies.
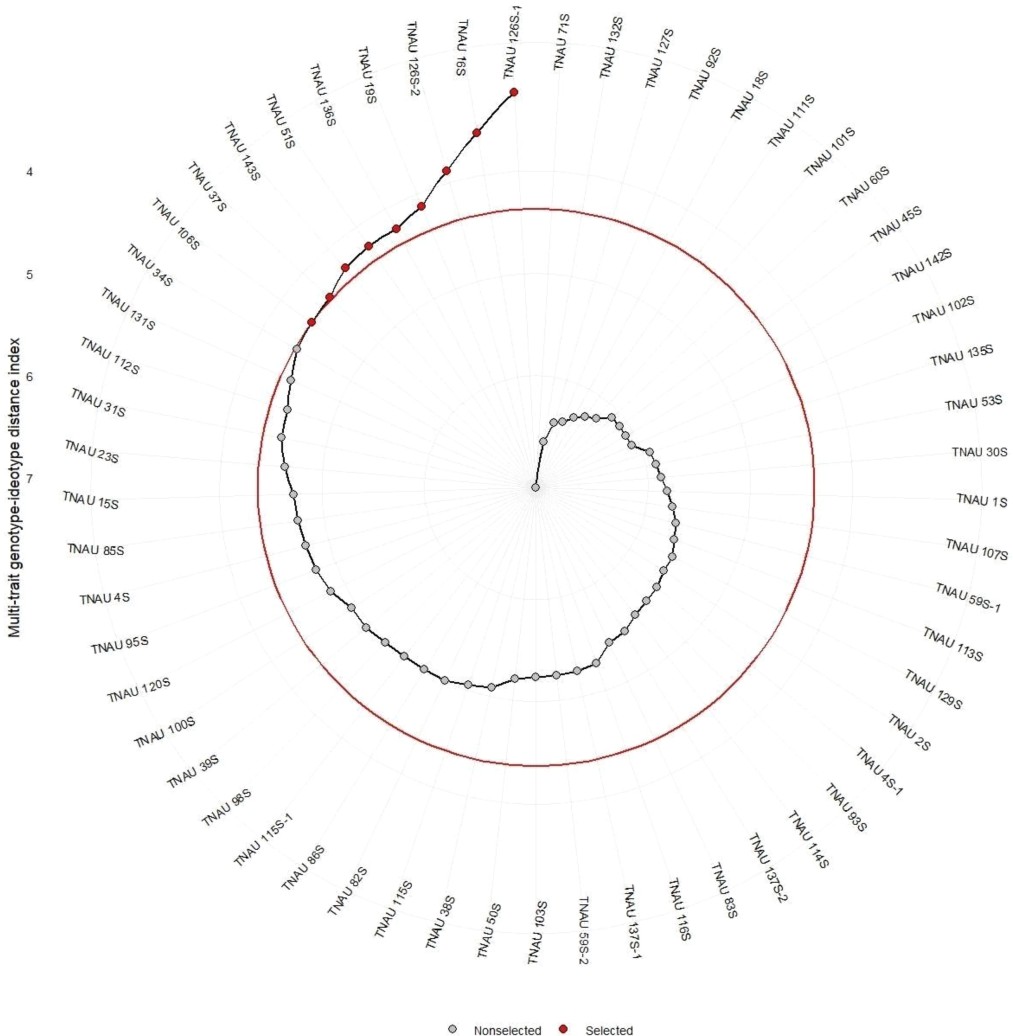

**Figure 4  Multi-trait genotype-ideotype distance index analysis for genotypic selection in rice.**

For marker RM81, the PIC value of 0.52 and heterozygosity index of 0.59 (Table 5) were the highest, signifying its utility in detecting polymorphisms and variability within rice germplasm populations. This observation aligns with findings from research assessing the genetic diversity and population structure in temperate japonica rice germplasm in Chile, which reported an average gene diversity of 0.52 across 30 SSR loci, underscoring the markers' effectiveness in genetic studies (*Gutiérrez et al., 2017*). The high PIC value of RM81 further emphasizes its potential for distinguishing genetic variations and aiding in the development of improved rice varieties.

The lower mean PIC (0.34) and heterozygosity index (0.28) observed in this study may also be attributed to the selection of random markers across all chromosomes, which may not adequately capture allelic variation specific to TGMS lines. Marker selection plays a crucial role in reflecting population-specific genetic diversity. As noted by *Panaud, Chen & McCouch (1996)*, marker choice can significantly influence diversity measurements, and

**Table 5** The list of polymorphic markers along with their number of alleles, polymorphic information content (PIC) value and heterozygosity index (H).

| S.no | Markers | H | PIC | S.no | Markers | H | PIC |
|------|---------|------|------|------|---------|------|------|
| 1 | RM319 | 0.21 | 0.19 | 23 | RM195 | 0.49 | 0.37 |
| 2 | RM22597 | 0.39 | 0.32 | 24 | RM5704 | 0.49 | 0.37 |
| 3 | RM3533 | 0.22 | 0.2 | 25 | RM184 | 0.28 | 0.24 |
| 4 | RM7653 | 0.19 | 0.17 | 26 | RM433 | 0.37 | 0.3 |
| 5 | RM4601 | 0.39 | 0.35 | 27 | RM423 | 0.47 | 0.36 |
| 6 | RM13912 | 0.21 | 0.2 | 28 | RM561 | 0.44 | 0.35 |
| 7 | RM36 | 0.25 | 0.23 | 29 | RM511 | 0.4 | 0.32 |
| 8 | RM81 | 0.59 | 0.52 | 30 | RM5931 | 0.19 | 0.17 |
| 9 | RM1018 | 0.5 | 0.37 | 31 | RM7403 | 0.29 | 0.25 |
| 10 | RM286 | 0.46 | 0.35 | 32 | RM349 | 0.21 | 0.19 |
| 11 | RM1896 | 0.47 | 0.36 | 33 | RM3917 | 0.37 | 0.3 |
| 12 | RM5709 | 0.32 | 0.27 | 34 | RM521 | 0.47 | 0.36 |
| 13 | RM310 | 0.48 | 0.36 | 35 | RM552 | 0.46 | 0.35 |
| 14 | RM559 | 0.32 | 0.27 | 36 | RM129 | 0.16 | 0.15 |
| 15 | RM16559 | 0.31 | 0.26 | 37 | RM335 | 0.37 | 0.3 |
| 16 | RM420 | 0.29 | 0.25 | 38 | RM457 | 0.19 | 0.17 |
| 17 | RM282 | 0.22 | 0.2 | 39 | RM427 | 0.42 | 0.33 |
| 18 | RM210 | 0.19 | 0.17 | 40 | RM205 | 0.39 | 0.31 |
| 19 | RM125 | 0.07 | 0.07 | 41 | RM8263 | 0.42 | 0.33 |
| 20 | RM5352 | 0.4 | 0.32 | 42 | RM1337 | 0.35 | 0.29 |
| 21 | RM4455 | 0.35 | 0.29 | 43 | RM346 | 0.47 | 0.36 |
| 22 | RM157 | 0.41 | 0.33 | | | | |

the random selection of markers in this study likely contributed to the lower observed PIC values. The reduced genetic variation is also a known outcome of selective breeding, where a specific genetic profile is emphasized to maintain desirable traits. The clustering analysis using Jaccard distances and unweighted pair group method with arithmetic mean (UPGMA) revealed eight distinct clusters (Fig. 5), with larger clusters such as I and II indicating relatively less divergence among genotypes within these groups. Smaller clusters, such as IV, V, and VI, suggest greater genetic variation among the genotypes included. The maximum Jaccard distances observed in this study 0.736 between TNAU131S and TNAU112S and 0.7121 between TNAU131S and TNAU34S align closely with the findings of *Shivani et al. (2021)*, who reported a maximum Jaccard distance of 0.7 among highly diverse rice genotypes. This consistency indicates that while the TGMS lines exhibit limited genetic diversity overall, certain genotypes within the group are substantially divergent.

The minimum Jaccard distance in this study, 0.20 between TNAU30S and TNAU98S, is comparable to the 0.22 minimum distance reported by *Seetharam, Thirumeni & Paramasivam (2009)*. This suggests that despite the selective breeding focus of TGMS lines, some genetic variability persists, potentially due to residual variation within the parental gene pool. *Sai Rekha et al. (2021)* similarly reported molecular diversity in six TGMS lines using 27 SSR markers, highlighting that even within highly specialized breeding

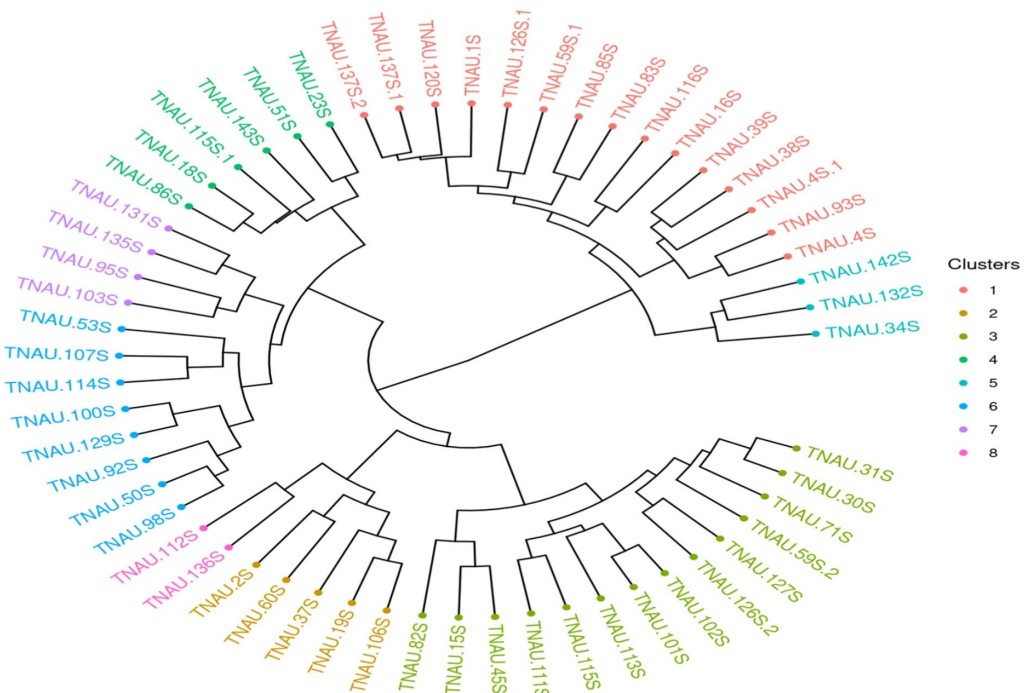

**Figure 5   UPGMA clustering of rice genotypes based on SSR marker data.**

populations, there remains a baseline level of genetic differentiation. The grouping of genotypes into clusters may also reflect deviations caused by the lower genome coverage of molecular markers used. As noted by *Seetharam, Thirumeni & Paramasivam (2009)*, low marker density can introduce sampling errors, leading to variations in cluster formation and influencing dendrogram interpretations. These deviations emphasize the importance of comprehensive genome coverage for accurately representing genetic relationships. The observed clustering pattern in this study is consistent with prior reports, where molecular markers with limited genome coverage lead to grouping errors due to under sampling.

## Population structure

The identification of two subpopulations within the 57 TGMS rice lines, supported by the Bayesian model-based STRUCTURE analysis, provides significant insights into genetic diversity and its implications for breeding. The Δk value of 120 confirmed the presence of two subpopulations, SG1 and SG2, with distinct genetic architectures (Fig. 6). This is consistent with findings by *Garris et al. (2005)* and *Parida et al. (2012)*, who reported similar sub-structuring in rice populations, driven by environmental selection and geographic isolation.

The analysis of molecular variance (AMOVA) in TGMS rice populations highlights significant genetic diversity. *Nachimuthu et al. (2015)* reported 14% variation between groups and 86% within groups. Similarly, the current AMOVA results showed 22% variation among populations and 78% among individuals, indicating genetic divergence

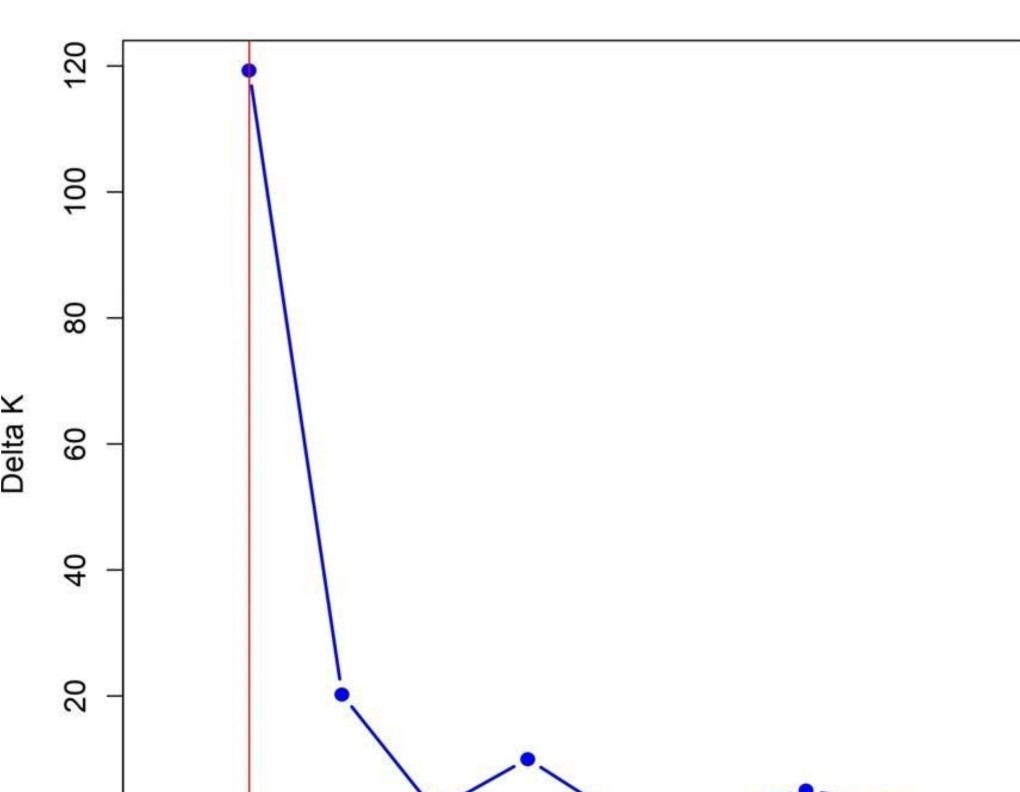

**Figure 6** Delta K analysis for determining optimal population structure clusters.

influenced by breeding objectives and substantial individual-level diversity essential for resilience and hybrid development (Table 6). The absence of variation within individuals (0%) reflects the homozygous nature of TGMS lines, vital for stability in sterility traits. An estimated variance of 9.189 underscores the importance of genetic diversity for hybrid vigor and adaptability in rice breeding programs.

SG1 exhibits higher admixture (28.5%), suggesting gene flow from multiple sources, while SG2 demonstrates greater genetic homogeneity (16.6% admixture), indicative of selective breeding and isolation (Fig. 7). These observations align with the work of *Choudhury et al. (2023)* and *Nie et al. (2022)*, who demonstrated that environmental adaptations and selection pressures significantly influence the genetic structure of rice populations. The clustering based on Jaccard distances and molecular markers highlights clear genetic differentiation between the two subpopulations. Molecular markers such as RM423 and RM521, which track alleles linked to adaptive traits, have proven effective
**Table 6  Analysis of molecular variance between subpopulations.**

| Source | Df | MS | Estimated variance | Percent variation |
|---|---|---|---|---|
| Among populations | 1 | 119.330 | 1.977 | 22% |
| Among individuals | 55 | 14.423 | 7.212 | 78% |
| Within individuals | 57 | 0.000 | 0.000 | 0% |
| Total | 113 | | 9.189 | 100% |

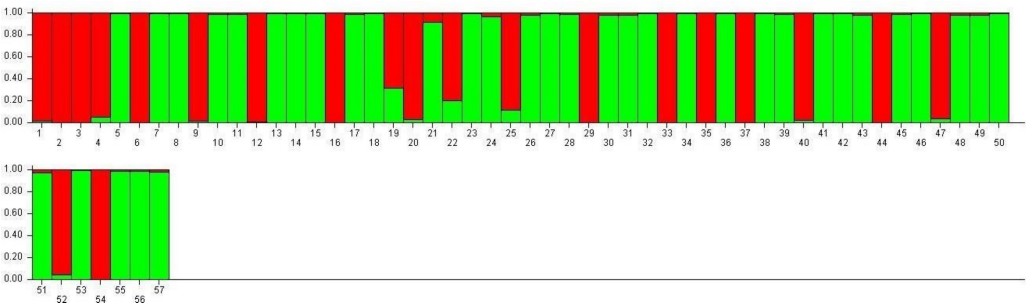

**Figure 7  Visualization of parental line allocation into distinct subgroups.**

**Table 7  Pairwise genetic differentiation (Fst) and gene flow (Nm) between three subpopulations along with observed (Ho) and expected heterozygosity (He).**

| | Pairwise genetic differentiation (Fst) and gene flow (Nm) | | | |
|---|---|---|---|---|
| | Population 1 | Population 2 | Ho | He |
| Population 1 | 0.000 | 0.215 (Fst) | 0.000 | 0.314 |
| Population 2 | 1.937 (Nm) | 0.000 | 0.000 | 0.329 |

in distinguishing genetic groups. This mirrors the findings of *Parida et al. (2012)*, who emphasized the importance of SSR markers in identifying subpopulation-specific traits. The moderate gene flow (Nm = 1.937) and genetic differentiation (Fst = 0.215) observed between SG1 and SG2 (Table 7) are consistent with *Yamasaki & Ideta's (2013)* conclusions that moderate differentiation (Fst values between 0.2 and 0.3) can facilitate heterosis by balancing shared and distinct genetic traits. The AMOVA results reveal 22% variation among populations and 78% within populations, underscoring the dominance of intra-population diversity. This balance is crucial for TGMS breeding to ensure genetic stability while optimizing hybrid seed production.

The high admixture in TNAU 53S, derived from GD 98049-29 selection, suggests deliberate efforts to integrate diverse genetic material for adaptability or enhanced trait expression. Similar strategies have been highlighted by *Rahman et al. (2011)*, who noted the benefits of leveraging gene flow to improve genetic diversity in breeding populations.

The low observed heterozygosity (Ho = 0.000) in TGMS populations is consistent with the findings of *Salem & Sallam (2016)*, who observed similar patterns in self-pollinating rice populations. However, the moderate expected heterozygosity values (He = 0.314 for SG1 and 0.329 for SG2) align with *Nachimuthu et al. (2015)*, suggesting sufficient genetic potential for variation within these populations. Discrepancies between molecular and morphological clustering in this study reflect differences in the genetic and phenotypic determinants of diversity. Molecular markers provide a clearer separation of genetic groups, while morphological traits are influenced by complex processes such as pleiotropy and epistasis, as highlighted by *Pathak et al. (2020)* and *Rahman et al. (2011)*. These processes can obscure clustering patterns, particularly in traits influenced by environmental factors. *Chen et al. (2024)* emphasized the importance of accounting for genotype-environment interactions in breeding programs targeting diverse agro-climatic zones, as environmental conditions significantly impact sterility and fertility traits in TGMS lines.

The clustering of TGMS lines into SG1 and SG2 has clear implications for breeding strategies. SG1's higher admixture and genetic diversity make it a valuable source for introducing novel traits, while SG2's genetic purity offers a stable base for hybrid seed production. Controlled cross-pollination, as suggested by *Salem & Sallam (2016)*, could facilitate the transfer of desirable traits between subpopulations, ensuring the development of robust TGMS lines suited to varied environments. The moderate Fst value (0.215) further supports the potential for gene introgression while maintaining critical sterility traits necessary for TGMS lines. The distinction between molecular and morphological clustering also highlights the importance of integrating diverse datasets to select superior parents for hybrid production. *Pathak et al. (2020)* and *Rahman et al. (2011)* observed similar differences, reinforcing the need for multifaceted approaches in breeding programs.

In conclusion, the results underscore the interplay between genetic structure, gene flow, and environmental adaptation in shaping the diversity of TGMS rice lines. The findings align with the broader literature, including *Garris et al. (2005)*, *Parida et al. (2012)*, and *Nachimuthu et al. (2015)*, demonstrating the potential of SG1 and SG2 for enhancing hybrid rice production. Leveraging the genetic potential of these subpopulations through targeted breeding strategies can ensure the development of stable and high-performing TGMS lines, capable of meeting the challenges of diverse agro-climatic conditions.

## CONCLUSION

This study highlights the pivotal role of TGMS lines in hybrid rice breeding, identifying traits such as stigma exertion, grains per panicle, and single-plant yield as key candidates for direct selection due to their high heritability and minimal environmental influence. Molecular analysis revealed unique genetic groups among TGMS lines *via* UPGMA clustering and PCA, with genotypes like TNAU 112S, TNAU 114S, TNAU 16S, and TNAU 136S showing exceptional promise for heterotic hybrid development. PCA underscored productive tillers, spikelet fertility, and panicle architecture as critical contributors to variability. The study aligns TGMS traits with current breeding priorities, including stress tolerance and climatic resilience, by emphasizing early blooming, compact architecture, and improved spikelet fertility for enhanced adaptation and stability. However, limitations

include the narrow environmental scope, limited SSR markers, and the omission of key traits like drought tolerance and pest resistance.

To maximize the utility of TGMS lines, future work should focus on multi-environment trials, integrate advanced tools like CRISPR and genome-wide association studies (GWAS), and emphasize stress-resilience traits. This research provides a solid foundation for developing high-yielding, climate-resilient rice varieties, addressing global food security challenges.

### Funding
This work was supported by ICAR-"Consortium Research Platform (CRP) on Hybrid Technology: RICE" and Department of Rice, Tamil Nadu Agricultural University. The funders had no role in study design, data collection and analysis, decision to publish, or preparation of the manuscript.

### Grant Disclosures
The following grant information was disclosed by the authors:
ICAR-"Consortium Research Platform (CRP) on Hybrid Technology: RICE" and Department of Rice, Tamil Nadu Agricultural University..

### Competing Interests
The authors declare there are no competing interests.

### Author Contributions
- B Nagendra Naidu conceived and designed the experiments, performed the experiments, analyzed the data, authored or reviewed drafts of the article, and approved the final draft.
- Manonmani Swaminathan conceived and designed the experiments, performed the experiments, analyzed the data, authored or reviewed drafts of the article, and approved the final draft.
- Pushpam Ramamoorthy conceived and designed the experiments, performed the experiments, analyzed the data, authored or reviewed drafts of the article, and approved the final draft.
- Kumaresan Dharmalingam conceived and designed the experiments, performed the experiments, analyzed the data, authored or reviewed drafts of the article, and approved the final draft.
- Raveendran Muthurajan conceived and designed the experiments, performed the experiments, analyzed the data, authored or reviewed drafts of the article, and approved the final draft.
- Selvi Duraisamy conceived and designed the experiments, performed the experiments, analyzed the data, prepared figures and/or tables, authored or reviewed drafts of the article, and approved the final draft.

- Nivedha Rakkimuthu conceived and designed the experiments, performed the experiments, analyzed the data, prepared figures and/or tables, authored or reviewed drafts of the article, and approved the final draft.
- Abirami Subramanian conceived and designed the experiments, performed the experiments, analyzed the data, prepared figures and/or tables, authored or reviewed drafts of the article, and approved the final draft.
- Rithesh Natarajan conceived and designed the experiments, performed the experiments, analyzed the data, prepared figures and/or tables, authored or reviewed drafts of the article, and approved the final draft.
- Bonipas Antony John conceived and designed the experiments, performed the experiments, analyzed the data, prepared figures and/or tables, authored or reviewed drafts of the article, and approved the final draft.

## Data Availability

The raw data is available in the Supplementary Files.

## Supplemental Information

Supplemental information for this article can be found online at http://dx.doi.org/10.7717/peerj.18975#supplemental-information.

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
