# Peer review of "Genotypic and phenotypic characterization of thermo-sensitive genic male sterile (TGMS) rice lines using simple sequence repeat (SSR) markers and population structure analysis"

_PeerJ, doi:10.7717/peerj.18975_

## Round 0.1 · original submission · Major Revisions

Dear Authors

The manuscript cannot be accepted for publication in its current form. It needs a major revision before publication. The authors are invited to revise the paper considering all the suggestions made by the reviewers. In addition, I believe that the manuscript should improve in terms of the following:

- Research questions and aim. I think that this is not clear in the text.
- Description of methods. Especially concerning statistical analyses, which appear to be poorly detailed.
- General structure. Figure citation, description, and discussion of results,...
- Conclusions. I believe this section should be improved.

Please note that the requested changes are required for publication.

With Thanks

·

Basic reporting

The article is clear and the well written.
The context is well explained.

However, I still have some comment/suggestions:

Abstract:
The second sentence, “These rice lines enhance genetic diversity by enabling hybrid seed production without manual emasculation, promoting greater trait recombination” is not in accordance with your explanation of the usual use of the CMS method for hybrid rice in the introduction -> hybrid rice are already available ? Also for the part “enhance genetic diversity”, this is true only within field if the hybrid rice is substituting a line …

L142-143: Except on chromosome 6
L171: The Anova presented in table 3 does not show that all genotypes are significantly different for all traits, but that some genotypes are significantly different for all traits, and thus there is a genotype effect for all traits.
Tables: The 43 markers listed on table 2 and table 7 are not all the same: RM205, RM8263, RM1337 and RM346 are not in the table 2, but are in table 7 and in the genotyping dataset.

Experimental design

The number of genotypes is relatively low and the strength of this article is to combine a many traits with a genotyping dataset, with many types of analysis.
It could be interesting to link a bit more analysis carried out on phenotypes and on genotypes.
Do you have any hypotheses about the origin of the 2 subpopulations that you could discuss here, and maybe link it with the phenotypes? Did you know of this population structure before the study? I saw that there are 2 markers on chromosome 2 (RM423 and RM521) totally differentiating both subpopulations …

Validity of the findings

The subject is important for rice breeding and the analysis described well the experiments.

Additional comments

Supplementary file with markers :

What mean the numbers at the top of the table ?

Why are each marker appearing twice ?

Supplementary file with phenotypes :

The names of the lines are not provided, are they in the same order than the file with genotypes ?

Reviewer 2 ·

Basic reporting

English. Some errors have been detected in the writing of English. These appear marked in the document.
 References and background. I think the background is correct. There is no uniformity observed in the list of bibliographic references.
 Article structure, figures, tables and raw data. I think the text, figures, and tables should be restructured according to the following comments:
o Lines 195-196. “The substantial genetic divergence between pairs such as TNAU 112S and TNAU 114S (308.51), and TNAU 112S and TNAU 113S (296.97),” Where was this information obtained from? from this research or from other authors? As written it is not well understood. In the abstract he talks about these genetic distances, however, the genetic analysis is later in the text. This can make it difficult for the reader to understand. If this information comes from this research, I think it is necessary to restructure the results and discussion section for a more fluid reading. If it is the result of other investigations, it can remain that way.
o Figures and Tables. No figure is cited in the text. Figures and tables must be cited appropriately when the results are described.

I think some results are missing. I think very important that authors include results obtained from the SSR markers in agarose gels. These figures could be part of a supplementary figure.

Experimental design

Aims and scope. I consider the manuscript to be suitable for the 'Plant Biology' section of the journal PeerJ.
 Research question. The research question and objective of the work do not appear clearly in the text. I think it would be good for the authors to review the document and specify these two issues more clearly.
 Material and Methods. Some points of material and methods should be reviewed and improved:
o Line 141. Composition of master mix.
o Lines 143-148. Please, review the protocol for SSR. Besides you should indicate that the annealing temperature for some primers is 50ºC.
o Line 138. Incluide the manufacturer.
o Line 151. Amplified base pairs. This information is lacking in the Table 2.
o Line 153. Please, include the reference for Jaccard distance method.
o Statistical analysis. Please, include detailed information about ANOVA, PCA, and so on. For example, p-value (only included in the figures or tables). This is one of the most important pairs in the manuscript, as they validate your findings.
o Lines 300-301. Bayesian model-based technique in STRUCTURE 2.3.4, with 50,000 burn-in iterations’ This information should be included in the material and methods section.

Validity of the findings

Data:
o Line 107. You indicate 57 in the text, but only 56 are indicated in the table.
o Line 139. Ideal by optimal.
o Lines 182-183. Keep the same numbers for PCV and GCV in the text as in the table.
o Line 204. (4.03) Where does this value come from? Please specify.
o Line 239. PCA. TNAU 136S seems to be closer to the rest than TNAU93S. Where is 92S?
o Table 7. Please, explain abbreviations.
o Line 276. Investigation by research.
 Conclusions. I think the biggest part of the conclusion could be the final paragraph of results and discussion. In my opinion, although the conclusions should have more prominent information about the results, they should focus on the importance and applicability of the results obtained in this research. However, I believe that the objective of the research is not clearly specified in the text and this makes a global understanding of this research difficult.

Additional comments

Dear authors,

Hope you are fine.
I send your manuscript with labels for faster review.

Annotated reviews are not available for download in order to protect the identity of reviewers who chose to remain anonymous.

Reviewer 3 ·

Basic reporting

The manuscript titled "Genotypic and Phenotypic Characterization of TGMS Rice Lines Using SSR Markers and Population Structure Analysis" provides valuable insights into the genetic and phenotypic diversity of TGMS (Thermosensitive Genic Male Sterile) rice lines. However, there are areas that need revision to enhance the scientific rigor, including a more in-depth discussion of statistical analyses and clearer articulation of key findings in relation to their broader implications for rice breeding. The manuscript would also benefit from addressing grammatical errors and improving the clarity of certain technical descriptions.
The abstract does not clearly articulate the objective of the study. It jumps into describing TGMS rice lines and their advantages but fails to state a concise research question or the primary aim of the study. Adding a sentence or two explicitly outlining the study’s goal would improve focus. There is no specificity regarding the experimental design. An abstract should link the specific results to wider implications, making it more compelling. Moreover, the abstract ends with a somewhat vague statement about "insightful information on genetic links and phenotypic associations," but it doesn’t provide a clear conclusion. Briefly, provide the results with key values.
In introduction, the use of outdated statistics from FAO (1996) to project rice demand in 2025, despite more recent data being available. Replace or complement the FAO (1996) citation with more current estimates from FAO or other authoritative bodies (e.g., FAOSTAT, IRRI). This would make the introduction more relevant to today’s context.
While the introduction mentions the advantages and drawbacks of both CMS and TGMS systems, it does not clearly define the specific gap in research that the current study addresses. Clarify what aspects of TGMS lines are still under-researched and how the present study addresses these gaps.
The literature on SSR markers lacks depth and justification for their use. It only mentions their polymorphism and co-dominant nature but does not explain why SSRs are particularly suitable for TGMS lines. Elaborate on the significance of SSR markers for genotypic characterization in rice breeding, especially in comparison to other available markers (e.g., SNPs, AFLPs). Also, explain why these markers are especially valuable for TGMS lines in identifying genetic diversity and ensuring the success of hybrid seed production.
While the introduction explains the TGMS system’s advantages, it does not sufficiently emphasize the significance of the two-line breeding system in improving hybrid vigor. Provide more detailed comparative insights into why the two-line breeding system with TGMS is preferable over the three-line CMS method.
Add a section addressing the environmental resilience of TGMS lines, including their adaptability to various agro-climatic zones, and how their sterility-fertility behavior may be affected by these stressors.
Reorganize the content for better logical flow. Start by discussing the general background of rice breeding, then move to CMS and its limitations, followed by TGMS and its advantages. After that, explain the role of molecular markers and finally justify the use of SSR markers in this context.
Although marker-assisted selection (MAS) is mentioned, its role in improving the efficiency of TGMS line development is underexplored. Discuss how MAS is accelerating breeding programs for TGMS lines, perhaps by providing examples of how MAS has already improved traits such as yield or stress resistance in rice. Tie this more directly into the study's objectives. Explicitly state the research objectives or hypotheses toward the end of the introduction.
In Material and Methods section, provide more detailed information on the genetic background and history of these TGMS lines. Specifically, mention the parental lines or geographic sources to help contextualize the population's diversity. If available, include references to earlier studies that have characterized these lines.
The description of field conditions and agronomic practices is vague. Terms like "standard recommendations" for fertilizer application are not specific enough for reproducibility. Clarify the specific environmental conditions (e.g., soil type, temperature, humidity) and provide detailed information on the amounts and types of fertilizers, irrigation schedules, and pest management protocols followed during the experiment.
While a randomized block design (RBD) with three replications is mentioned, the actual sample size for each genotype is not explicitly stated. Specify how many individual plants per genotype were used for data collection. Include the total number of plants evaluated across all repetitions and provide the rationale for the selected sample size.
For spikelet fertility, indicate the exact method used to assess sterility and fertility at the reproductive stage.
Mention whether agarose gel electrophoresis was used to check DNA integrity. Furthermore, details about the PCR conditions could be enhanced by specifying the Tm (melting temperature) for primer annealing, the number of cycles used, and how optimization was performed for SSR primers.
While 43 SSR markers were used, there is no explanation for their selection. It is unclear whether these markers were previously validated for TGMS rice or if they were chosen based on genomic coverage. Provide a justification for the choice of SSR markers. Explain whether they are known to be linked to fertility genes in rice or if they were chosen for their polymorphism across different rice genotypes. Also, clarify if markers covering important chromosomal regions related to fertility and agronomic traits were prioritized.
Describe the criteria for scoring SSR markers in more detail. Explain how you dealt with potential artifacts or weak amplifications. Also, clarify whether all 43 SSR markers were polymorphic or if any were excluded from the analysis due to poor amplification.
The Results and Discussion section feels cluttered due to the lack of clear separation between results and interpretation. A more structured approach, where results are clearly presented first, followed by a thorough interpretation in the discussion section, would enhance readability. The combination of both result presentation and discussion, making it hard for readers to differentiate between empirical findings and theoretical interpretation.
The mention of significant differences between genotypes is made, but no actual p-values or F-statistics are provided. Including these statistical values would add credibility to the analysis, allowing readers to understand the strength of these differences.
In the genetic variability subsection, the interpretation of Phenotypic and Genotypic Coefficient of Variation (PCV & GCV) could be expanded. Merely stating that PCV is higher than GCV, indicating environmental influence, does not provide enough depth. The magnitude of this difference and its implications for specific traits need to be addressed further. For instance, what percentage of variance is environmentally driven, and what is the practical implication for breeding programs?
The presentation of heritability and genetic advance values is informative, but further explanation of their biological significance would strengthen the discussion.
The Principal Component Analysis (PCA) results are presented well, but there are missed opportunities for in-depth interpretation. While the variance explained by each principal component (PC) is outlined, the contribution of specific traits to each PC is superficially mentioned. The discussion could be enhanced by explaining how the relationships between traits (e.g., the inverse association between plant height and pollen fertility) impact breeding decisions. Link the PCA findings to specific breeding outcomes, especially with TGMS lines that have potential for improving yield-related traits.
The MTSI analysis is another area where the results are under-discussed. Although the TGMS lines with superior performance and stability are identified, the rationale behind their selection and the breeding implications of these traits are not adequately explained. There is a need for a clearer justification of how these lines were selected and how their stability contributes to hybrid rice improvement.
The comparison with previous studies (e.g., Azam et al., 2023) lacks detail. More specific context about how this study's findings align or diverge from existing literature would help readers gauge the novelty and reliability of the results.
When discussing polymorphic information content (PIC), it is stated that the values are lower than those reported by earlier studies. This discrepancy needs more detailed interpretation—why might this study have found lower PIC values? Are the SSR markers used suboptimal for this population?
The UPGMA clustering results are mentioned, but the implications of the genetic distances between clusters are not fully explored. How does this clustering relate to the practical breeding of TGMS lines? Furthermore, the clustering of morphological and molecular data is briefly discussed, but the differences between these clustering methods are not well analyzed. The authors mention environmental factors but do not explore how these could affect hybrid selection.
The Bayesian model-based approach used to identify subpopulations (SG1 and SG2) provides interesting results, but more interpretation is needed. While the article notes that genetic divergence between the subpopulations is significant, there is no discussion of how this genetic structure impacts breeding strategies. For example, what breeding approaches might be best suited for crossing individuals from SG1 and SG2?
The analysis of gene flow and genetic divergence (Fst values) is interesting but lacks a clear biological context. What practical consequences do these divergence levels have for breeding efforts? Additionally, the discussion on heterozygosity values could be expanded to include more insights into how these findings align with the reproductive behavior of rice.
The conclusion lacks a broader discussion on how these findings contribute to the overall field of rice breeding and hybridization. There is no acknowledgment of any limitations of the study, please provide limitations. Additionally, the conclusion doesn't explain why the studied morphological and molecular traits are advantageous in modern rice breeding programs or how they align with existing breeding goals, such as climate resilience or stress tolerance. The conclusion does not suggest future research directions, please provide.

Experimental design

Experimental design is not clear. Although the planting spacing of 20 cm × 20 cm is provided, it would be beneficial to include the plot size and block layout (i.e., how the TGMS lines were distributed within the randomised blocks), which is essential to fully understand how the experimental area was structured. While it is mentioned that "standard recommendations" for fertiliser and intercultural techniques were followed, the exact fertiliser application rates (NPK ratios, timing, etc.), irrigation practices, and any pest or disease management strategies are not explicitly provided. These details are necessary for reproducibility.

Validity of the findings

The findings are promising; however, the authors should provide a more in-depth interpretation of data and technical discussion of the results.

---

## Round 0.2 · Minor Revisions

Dear Authors

The manuscript still needs a minor revision before publication. The authors are invited to revise the paper considering all the suggestions made by the reviewers. Please note that the requested changes are required for publication.

With Thanks

·

Basic reporting

This paper aims at describing the phenotypic and genetic diversity of TGMS rice lines. The paper has been greatly improved.

Experimental design

The experimental set up and results are well described.
However, a description of the method used for Genetic Advance as a Percentage of Mean is missing from the material & methods.

Validity of the findings

The results and conclusions of this paper will be valuable for hybrid rice breeding.
L520: “High performing” : not sure how to see this

Additional comments

Supplementary tables : Add titles to the tables

Reviewer 2 ·

Basic reporting

- Figures and Tables. Please, change Figure 5 by Fig. 5 (line 497tracked changes).

Experimental design

No comments.

Validity of the findings

- Line 151. Amplified base pairs. This information is lacking in the Table 2.
Your response: Thank you for your observation. The amplified base pair information has been included in Table 2 as suggested.
I could not find this in the new Table 2, however, I found it in the Suppl. Table 2. In any case, I suggest to cite supplementary figures and tables in the text.

Additional comments

No comments.

Reviewer 3 ·

Basic reporting

I appreciate the authors' efforts in addressing the majority of the queries raised in the previous round of revision. However, there are still a few remaining issues that require further attention.
Grammatical mistakes exist in the manuscript like in L70-L71 “Ensuring the timing of planting and site selection is crucial to maintain sterility and produce high-quality hybrid seeds.” Please correct all such mistakes.
Please insert correct symbols e.g., for degree authors have used 0.
Mention acronym first like CSP and then use their abbreviations.
Add brief detail of experimental design in abstract section.
In the Introduction section, lines 79-94, the authors have cited information from the 1990s, which appears outdated. It is essential to include recent advancements in TGMS technology to provide a comprehensive and up-to-date context for the study.
Line No. 160 and 165, Please add formulas by using equation mode.
In results and discussion section, results are superficially discussed, please discuss your results technically with in-depth logical reasoning in the light of current literature.
The UPGMA clustering in the figures requires correction, as several edge values are truncated or cut off, which hampers proper interpretation. Please ensure that the complete values are clearly visible and appropriately placed within the figures.

Experimental design

Experimental design is appropriately described.

Validity of the findings

Findings seems promosing.

---

## Round 0.3 · Minor Revisions

Dear Authors
The manuscript still needs a minor revision before publication. The authors are invited to revise the paper considering all the suggestions made by the reviewers. Please note that the requested changes are required for publication.
With Thanks

·

Basic reporting

The authors considered well the comments made by all the reviewers and the paper was much improved. This research will contribute to the rice breeding community.

Experimental design

MTSI : defined as Multi-Trait Stability Index L349, as Multi-Trait Selection Index L350, and as Multi-Trait Genotype-Ideotype Distance Index in the title of the figure 4. It would be helpful to add some informations on the MTSI analysis in the material and methods and the purpose of the analysis.

Validity of the findings

The results are well described.

Additional comments

In the conclusion L571: High performing : how to see this?

Reviewer 2 ·

Basic reporting

No comments.

Experimental design

No comments.

Validity of the findings

No comments.

Additional comments

No comments.

Reviewer 3 ·

Basic reporting

Authors tried to answer the previous queries but still certain improvements are required, as the results and discussion and conclusion sections are overly detailed, and lacks a clear logical flow. Certain points are repeated, which affects readability. Remove repetitions.
Over-reliance on presenting detailed statistical values (e.g., PCV, GCV, heritability, GAM) for each trait interrupts the narrative flow. Condense the detailed statistics and focus on summarizing the most critical traits that directly contribute to hybrid rice breeding objectives.
The discussion repeats information already provided in the introduction or methods (e.g., the importance of TGMS lines in hybrid rice breeding). Streamline the content by avoiding repetition and focusing on interpreting the results in the context of the study objectives.
While the results are statistically significant, the biological implications of some traits (e.g., stigma length and glume angle) could be explained more clearly. For example, explain how traits like high stigma exertion specifically enhance outcrossing rates or hybrid vigor.
The integration of previous studies (e.g., Kumar et al., 2016; Lopez et al., 2000) is useful but sometimes excessive and distracts from the primary findings. Use references selectively to support key findings, avoiding overloading the text with citations.
Explain how crossing genetically distinct lines like TNAU 112S and TNAU 114S can be strategically implemented in breeding programs to optimize heterosis.
The conclusion section includes excessive detail that detracts from its clarity and impact. For instance, the discussion on specific genotypes and their traits could be summarized succinctly. I recommend focusing on the key findings and their implications for hybrid rice breeding, ensuring the conclusion is appropriately aligned with the study objectives. Avoid reiterating detailed results or introducing new concepts, and instead emphasize the broader significance of your findings in a succinct and impactful manner.

Experimental design

Experimental design seems appropriate.

Validity of the findings

Findings seems promising.

---

## Round 0.4 · accepted · Accept

Dear Authors,

I am pleased to inform you that the manuscript has improved after the last revision and can be accepted for publication.

Congratulations on accepting your manuscript, and thank you for your interest in submitting your work to PeerJ.

With Thanks

·

Basic reporting

The paper is clear and was improved during the reviewing process.

Experimental design

The experimental design is well described.

Validity of the findings

The findings will be interesting for the rice breeding community.

Reviewer 3 ·

Basic reporting

The authors have satisfactorily addressed most of the suggestions and concerns raised in the previous round of revision.

Experimental design

The experimental design is well-structured and appears to be reproducible.

Validity of the findings

Findings seems valid.